# TAT-Modified ω-Conotoxin MVIIA for Crossing the Blood-Brain Barrier

**DOI:** 10.3390/md17050286

**Published:** 2019-05-12

**Authors:** Shuo Yu, Yumeng Li, Jinqin Chen, Yue Zhang, Xinling Tao, Qiuyun Dai, Yutian Wang, Shupeng Li, Mingxin Dong

**Affiliations:** 1Institute of Neuroregeneration & Neurorehabilitation, Qingdao University, 308 Ningxia Street, Qingdao 266021, China; 2Department of Protein Engineering, Beijing Institute of Biotechnology, Beijing 100071, China; o_yys@163.com (S.Y.); chenjq0210@163.com (J.C.); zy570524967@163.com (Y.Z.); daiqy@mail.bmi.ac.cn (Q.D.); 3State Key Laboratory of Oncogenomics, School of Chemical Biology and Biotechnology, Peking University Shenzhen Graduate School, Shenzhen 518055, China; wendyleeym@163.com (Y.L.); taoxinling_hqu@163.com (X.T.); 4Djavad Mowafaghian Centre for Brain Health and Department of Medicine, University of British Columbia, Vancouver, BC V5Z 1M9, Canada; ytwang@brain.ubc.ca; 5Campbell Research Institute, Centre for Addiction and Mental Health, Toronto, ON M5T 1R8, Canada; 6Department of Psychiatry, University of Toronto, Toronto, ON M5T 1R8, Canada

**Keywords:** ziconotide, TAT (the transactivator of transcription domain), peptide, analgesics, BBB (blood-brain barrier) penetration

## Abstract

As the first in a new class of non-opioid drugs, ω-Conotoxin MVIIA was approved for the management of severe chronic pains in patients who are unresponsive to opioid therapy. Unfortunately, clinical application of MVIIA is severely limited due to its poor ability to penetrate the blood-brain barrier (BBB), reaching the central nervous system (CNS). In the present study, we have attempted to increase MVIIA’s ability to cross the BBB via a fusion protein strategy. Our results showed that when the TAT-transducing domain was fused to the MVIIA C-terminal with a linker of varied numbers of glycine, the MVIIA-TAT fusion peptide exhibited remarkable ability to cross the bio-membranes. Most importantly, both intravenous and intranasal administrations of MVIIA-TAT in vivo showed therapeutic efficacy of analgesia. Compared to the analgesic effects of intracerebral administration of the nascent MVIIA, these systemic administrations of MVIIA-TAT require higher doses, but have much prolonged effects. Taken together, our results showed that TAT conjugation of MVIIA not only enables its peripheral administration, but also maintains its analgesic efficiency with a prolonged effective time window. Intranasal administration also rendered the MVIIA-TAT advantages of easy applications with potentially reduced side effects. Our results may present an alternative strategy to improve the CNS accessibility for neural active peptides.

## 1. Introduction

MVIIA (ziconotide) is an ω-conotoxin first isolated from the venom of Conus magus [1]. It contains 25 amino acids, six of which are cysteine residues that are linked in pairs by three disulphide bonds. MVIIA is a selective and reversible N-type calcium channel (Ca_v_2.2) inhibitor, and has been shown to be an effective anti-hyperalgesic. In animal models, intrathecal administration of MVIIA exhibited significant antinociceptive effect that is 100 times greater potency than that of intrathecal morphine [2]. As such, MVIIA was approved by the FDA and EMEA as the first in a new class of non-opioid drugs for the management of severe chronic pains in patients who were unresponsive to opioid therapy [3]. Unfortunately, due to its poor blood–brain barrier penetrating ability, it must be given as an intrathecal injection through an infusion pump [3]. The appended cost of infusion pump and the poor compliance of patients with invasive intrathecal administration tremendously restrict the clinical application of MVIIA. At present, MVIIA is mainly prescribed for long-term or permanent therapy of chronic pain states, and management of refractory to existing, more conservative pain.

To explore the plausibility of circumventing direct MVIIA delivery into the cerebrospinal fluid (CSF), numerous attempts have been reported. Anand P and coworkers developed a bi-functional viral nanocontainer with MVIIA in the interior cavity, and demonstrated its ability to successfully transport MVIIA in several BBBmodels of rat and human [4]. However, whether MVIIA could be delivered to CSFvia intravenous administration and produces analgesic effects in animals is yet unknown. Murthy et al. investigated the efficacy of MVIIA administered intranasally either in the form of solution or Kolliphor P 407 gels (KP 407) in Sprague–Dawley rats [5]. In the group receiving ziconotide with chitosan, the bioavailability of MVIIA in CSF following intranasal administration was only two fold greater than the bioavailability following intravenous administration, which is far from the efficacy requirement of drug development [5].

As 98% small molecules and 100% for large molecules could not penetrate the BBB, drug translocation across the BBB remains a formidable challenging task. Except for invasive procedures, numerous attempts to breach the BBB have been explored, and these attempts include making chemical modifications to improve the cargo molecules’ pharmacoketic and pharmacodynamic characteristics, and designing chimeric molecules by taking advantage of the physical transporter and receptor transcytosis systems that are employed for brain nutrient absorption [6]. However, these procedures also suffer from the limitation of applicable conditions, saturable characteristics of endothelial receptor and transporters, interfering physical translocation due to its competitive binding in nature, increased molecular size, and accompanied increase in cost [7]. On the other hand, an alternative procedure of cell-penetrating peptides (CPPs) represents a ubiquitously applicable transporting system for a variety of substances including peptides, RNA/DNA molecules, macromolecules such as antibodies and heterogenous nanoparticles [8].

Since the discovery of TAT, the transactivator of transcription domain encoded by human immunodeficiency virus type-1 (HIV-1), as a cationic protein that is able to enter cells [9], a vast number of CPPs from heterogeneous sources were found with distinct physicochemical properties, including TAT and penetratin, chimeric peptides like transportan, and synthetic peptides [10]. Most CPPs have the character of 10–30 amino acids in length, usually contain several basic amino acids such as arginine and lysine, allowing their electrostatic interactions with the negatively charged molecules exposed on the surface of cellular membranes like glycosaminoglycans (GAGs) and sialic acid residues, as well as rendering them water soluble and cationic or amphipathic in nature [11]. As a promising drug delivery tool, CPP presents several unique advantages compared to viral or nanoparticles. It is relatively safe and highly efficient, and could be versatility applied in various scenarios, including tumor targeting, vaccine delivery, and tumor diagnosis. More recently, the safety and efficiency to deliver active peptides into brain by Tat sequence were demonstrated in several clinical trials including NA-1 (Tat-NR2B9c), D-JNKI-1, and KAI-1678 [8].

In this study, we examined whether the fusion protein strategy with TAT peptide could overcome the stringent sequence and structure requirement of MVIIA for its analgesic efficacy. A serial of cationic TAT-modified peptide MVIIAs were designed and synthesized, and the resulting conjugates were evaluated for their in vitro and in vivo activity via peripheral, systemic administration. Both in vitro and in vivo results showed that C-terminal fusion with TAT peptide render its BBB penetration capacity, without significantly compromising its analgesic efficiency.

## 2. Results

### 2.1. Chemical Identity of MVIIA and its TAT-MVIIA Variants

After being treated for 24–48 h at 4 °C in the buffer containing 1 mM GSH, 0.1 mM GSSG, 1 mM EDTA, and 0.2 mg/mL peptide (pH 7.9), a linear peptide folding with one major peak and several minor Peaks were observed with HPLC analysis. The major products were purified and assessed with analytical reversed phase HPLC, and the purity of the peptides was determined as >98% (Scheme 1). All peptides had the expected molecular weights (Appendix A) ascertained by UltraflexIII TOF/TOF mass spectrometry (Bruker).

### 2.2. CD Spectroscopy

As shown in Figure 1, MVIIA presents a typical signature for β-sheet with a maximum near 195 nm and a minimum at 205 nm, but TAT variants showed a similar random coil structure with a negative band at ~200 nm, suggesting the secondary structure of the peptides remains unchanged when the length of the linker between MVIIA and TAT sequence expands. The molar ellipticity of TAT variants was deeper when linker expands, suggesting that the expansion of the linker between MVIIA and TAT sequence helped to form a random coil.

### 2.3. Inhibitory Effects on Ca_V_2.2 Channel Currents Induced by MVIIA and Its Variants

It is well known that MVIIA is a selective Ca_V_2.2 channel blocker. The inhibitory effect of 2 μM MVIIA on Ca_V_2.2 channel was more than 90%. In this study, we recorded the peak Ca^2+^ current (ICa) of Ca_V_2.2 channels (α1B, α2δ1, and β3) overexpressed in HEK 293T cells. All currents were evoked by a 100 ms voltage step from a holding potential between −80 and +10 mV. A bath application of 1 μM MVIIA, MVIIA-a, MVIIA-b, MVIIA-c, and MVIIA-d reduced ICa amplitude by 98.24 ± 0.708%, 89.45 ± 0.752%, 91.70 ± 1.477%, 98.81 ± 0.427%, and 84.26 ± 3.127%, respectively (n=5) (Figure 2A–G). MVIIA and MVIIA-c showed similar ability in blocking Ca_V_2.2 channel, while others showed a slightly lower inhibitory effect on Ca_V_2.2 channel currents. In contrary, l-MVIIA showed a significantly reduced ability in blocking Ca_V_2.2, with reduced ICa amplitude of 23.28 ± 3.347% at a concentration of 10 μM (Scheme 1). The concentration-response relationship for MVIIA inhibition of Ca_V_2.2 has an IC_50_ of 0.0436 μM (Figure 2A). Whereas the IC_50_ of TAT variants MVIIA-a, MVIIA-b, MVIIA-c, and MVIIA-d was 0.413, 0.379, 0.237, and 0.345 μM, respectively (Figure 2B–G). Thus approximately 5 to 10-fold reduction of IC_50_ of TATvariants in comparison with that of MVIIA suggests that the length of the linker between MVIIA and TAT sequence can affect the binding ability to Ca_V_2.2.

### 2.4. The Antinociceptive Effects of MVIIA and MVIIA-c

Since TAT-fused peptides have previously shown to have BBB-penetrating ability to render the fused peptides with CNS activity following a peripheral, systemic administration, we predict that TAT variants of MVIIA would have antinociceptive effects following systemic application. We next tested this prediction using a peripheral administration of MVIIA-c, the TAT variant of MVIIA with a double Gly linker that shed the best inhibitory effect on Ca_V_2.2 channel currents among all TAT variants tested. MVIIA and saline were used as positive and vehicle controls, respectively.

#### 2.4.1. The Antinociceptive Effect in the Hot-Plate Test

The antinociceptive effects of MVIIA and MVIIA-c were first examined in the hot-plate test. As shown in Figure 3A, MVIIA exhibited a maximal effect 1 h after administration (I.C.V.) at doses of 0.11, 0.33, or 1.00 nmol/kg, and the effect disappeared within 4 h after administration at dose of 1.00 nmol/kg. But MVIIA showed no effect when administered via intravenously administrations at any doses (Figure 3B). MVIIA-c, the TAT mutant of MVIIA, exhibited a maximal effect 3 h after administration at a dose of 1.00 μmol/kg via intravenously administration (Figure 3C). Interestingly, the analgesic effects of MVIIA-c appeared to last much longer time than that of MVIIA with the maximal effect lasting about 4 hours and disappearing within 12 h after the administration. Intranasal administration of MVIIA-c in a lower dose (9.9 nmol/kg) reached maximal analgesic effect in 0.5 h, and the level of analgesic effects was similar to that obtained with the intravenous administration of 0.33 μmol/kg of MVIIA-c (Figure 3D).

#### 2.4.2. The Antinociceptive Effect in Acetic Acid-Induced Writhing Test

MVIIA-c decreased the acetic acid-induced writhing numbers in a dose-dependent manner. The I.C.V. treatment with MVIIA-c at the dose of 0.6, 1.8, and 5.4 nmol/kg led to 14.75%, 39.53%, and 81.77% decrease in the acetic acid-induced writhing response compare to the control group of mice, respectively (Figure 4A). The I.V. treatment with MVIIA-c at the dose of 0.33, 1.00, and 3.00 μmol/kg led to 13.44%, 19.10%, and 52.83% decrease in the acetic acid-induced writhing response compare to the control group of mice, respectively (Figure 4B). In comparison, MVIIA, only when given intracerebroventricularly (0.11, 0.33, and 1.00 nmol/kg, I.C.V.), exhibited a reduction of writhing response by 8.97%, 53.37%, and 76.88%, respectively (Figure 4A), and it failed to attenuate writhing numbers when it was intravenously administrated (Figure 4B).

### 2.5. Effects of Peptides on the Coordinated Locomotion

Motor disorders and abnormalities in the nervous system is one of the typical side effects of MVIIA or MVIIA-c. Rotarod test was performed to evaluate the effects of these peptides (0.9 nmol/kg) on the coordinated locomotion function. Thirty minutes post-injection (I.C.V), the stay time was recorded as 156.1 ± 9.635, 74.5 ± 20.28, and 75.25 ± 13.8 s for saline, MVIIA and MVIIA-c (Figure 5A), respectively. However, MVIIA-c showed significantly effects on coordinated locomotion as compared with MVIIA at 120 min post-injection (Figure 5B).

### 2.6. Tremor Symptom in Mice Induced by Peptides

Tremor time is regarded as a typical side-effect symptom for MVIIA or MVIIA-c. In this study, the dose of 0.9 nmol/kg was selected for exploration. In the Figure 6, 30 and 120 min after I.C.V. administration, the accumulative tremor time of mice induced by MVIIA was 252.7 ± 28.64 and 227.1 ± 31.5, and by MVIIA-c was 202.8 ± 30.27 and 236.3 ± 22.02 s, respectively. MVIIA-c did not induce more severe tremor symptom than MVIIA at both 30 min and 120 min post-injection.

## 3. Discussion

Naturally biological peptides represent a large underexplored resource to develop peptide therapeutics and the analgesic drug ziconotide is an excellent example among them. Derived from the cone snail, it acts as a selective inhibitor of N type voltage-gated calcium channel [2]. It is estimated that the potency of MVIIA is 100 times higher than morphine [12]. However, apart from their distinguished advantages such as lower toxicity, fewer concerns of drug to drug cross-reactions, and less tissue accumulation, some of the disadvantages are prominent as well, including low bioavailability and stability, difficult for administration, delivery and storage [13]. Delivery of these therapeutic peptides to their targets in the central nervous system remains as a taunting challenge. As a result, MVIIA is only given intrathecally and prescribed when morphine tolerance happened.

Our results found that MVIIA-c showed a high potency as a N-type calcium channel blocker, with five times less activity than MVIIA, supporting the previous assumptions that the activity of MVIIA requires a strict stereo steric configuration. Previous nuclear magnetic resonance (NMR) spectroscopy revealed the triple-stranded anti-parallel β-sheet structure of MVIIA [14]. Six cysteine residues are linked by three pairs of disulphide bonds, that serve to stabilize the conformation and form four loops [15]. Although the detailed molecular structural mediating the binding of MVIIA to the N-type calcium channel is yet unknown, it is believed that the relative positions of amino acid side chains, but not the cystines, determine its potency. Derivatives with similar bioactivities to MVIIA should have similar secondary structures theoretically, so we first used circular dichroism as a simple characterization to determine the secondary structures of MVIIA derivatives. Unfortunately, the derivatives of MVIIA, whether TAT sequence is added to the N-terminal or C-terminal of MVIIA, do not show the expected secondary structure. Instead, in an agreement with the literature [16,17], they show random coil structure with a strong negative band at around 200 nm (Figure 1). The random coil CD signal with positive charged TAT sequence completely covers the secondary structure information of β-sheet that MVIIA derivatives should have, so it is difficult to be used to assess the bioactivities of the MVIIA derivatives.

MVIIA exerts its analgesic effects via the blocking of N-type calcium channels. We therefore evaluated the in vitro electrophysiological activity of MVIIA derivatives against N-type calcium channels. We attempted to add five amino acids to the N-terminal of MVIIA and found that the electrophysiological activity of the resulting peptide L-MVIIA was lost, which indicates that the biological activity of MVIIA is very sensitive to N-terminal modification. As such, we added the TAT sequence to the C-terminal of MVIIA and gapped with various numbers of amino acids linker. The derived fusion peptide MVIIA-a-d maintained its inhibitory activity toward N-type calcium channels, albeit with a slightly reduced efficacy in comparison with that of MVIIA. Activities of the fusion peptide MVIIA-a-d are dependent on the length of linker amino acids. When linker length is less than two amino acids, the bioactivity of peptides increases with the increase of linker length. However, when linker is three amino acids in length, the activity of peptide MVIIA-d is reduced again, lower than MVIIA-c with two amino acids. Thus, the bioactivity of MVIIA-c is the strongest among the synthesized derivatives (Figure 1). Interestingly, the hill slope increases with the increase of linker length, when linker is three amino acids in length the hill slope reaches to the max (2.387), while the hill slope of MVIIA is only 1.296. After that, the hill slope decreases again when the linker’s length continues to grow. As is known to us, hill slope reflects the binding situation of ligand and receptor. Since the linkage between inhibitor binding and response can be very complex, the mechanism of the change of hill slope in this case needs consideration in future studies. Although it is certainly possible that the MVIIA derivatives have the same disulfide bond pairing as MVIIA, the derivatives still display 5–10 fold less activity. This may also be caused by disulfide bond mismatch. Next, we will study the disulfide bond pairing of derivatives by digestion and mass spectrometry.

It is reported that when administered via intravenous route (100 μg/kg) in Sprague–Dawley rats, the C_max_ and T_max_ of MVIIA in CSF were found to be 37.78 ± 6.8 ng/mL and ~2 h, respectively. The AUC_0-6_ of MVIIA in CSF following i.v. administration was 4.98 ± 0.97 min.μg/mL (100 μg/kg dose). The poor bioavailability of drug in CSF following intravenous administration is likely due to the poor ability of MVIIA to cross the blood–CSF barrier which is an interface formed by the epithelial cells of the choroidplexus [5]. In order to test the ability of the TAT Modified MVIIA to penetrate cell membrane and BBB, we tested the analgesic activity of MVIIA-c on two classical analgesic mice models by intravenous administration. MVIIA showed no effect when intravenously administration at any doses in the hot-plate test. In great contrast, intravenous administration of MVIIA-c at a dose of 3.00 μmol/kg exhibited the same maximal analgesic effect as intrathecal administration of MVIIA at a dose of 1.0 nmol/kg. Interestingly, the maximal effect lasts about 4 hours and disappears within 12 h after administration (Figure 3). T_max_ of MVIIA-c in CSF were found to be ~2 h, which is same as intravenous administration of MVIIA, but slower than intrathecal administration of MVIIA (0.5 h). The analgesic effect of the same dosage was also observed in acetic acid writhing mice. Intrathecal administration MVIIA-c at five times the dose of MVIIA could achieve similar analgesic effect to that of MVIIA, which was consistent with the electrophysiological activity data (Figure 4A). But the total writhing number of intravenous administration of MVIIA-c were significantly decreased compared to that of MVIIA at the dose of 1.00 and 3.00 μmol/kg (Figure 4B). To our knowledge, this is the first time that the modified MVIIA has been shown to exhibit analgesic activity in an animal model by intravenous administration.

Although intravenous administration of MVIIA-c produces analgesic effects similar to that of intrathecal administration MVIIA, the required dose of MVIIA-c is much greater than that of intrathecal administration of MVIIA. In addition, systemic administration of ziconotide is also known to cause profound side effects [18]. These two problems may limit the potential intravenous application of MVIIA-c. Intranasal delivery is a noninvasive route that offers a direct pathway from nose to CSF via theolfactory apparatus [19]. Nose to CSF pathway could deliver drugs directly to the CSF bypassing the blood–CSF barrier. Indeed, we were able to show that analgesic activity of MVIIA-c on the hot-plate mice models by intranasal administration. As shown in Figure 3D, Intranasal administration of MVIIA-c can achieve the same speed analgesic activity as intrathecal MVIIA, T_max_ of MVIIA-c in CSF were found to be only 0.5 h. In order to achieve the same analgesic efficacy, the dose of intranasal administration MVIIA-c was only ten times that of intrathecal MVIIA. The more exciting thing was that the efficacy curve of MVIIA-c is smoother and longer time than that of MVIIA, which is consistent with intravenous administration MVIIA-c. These advantages may be explained by the facts that crossing the blood–brain barrier is a dynamic process and as such, MVIIA-c is in a relatively slow degradation environment. Once MVIIA-c crosses the cell membrane and enters the cerebrospinal fluid, the intermediate peptides remain bioactive even if part or all of the TAT sequence is digested by enzymatic digestion. TAT has been widely used as a cell-penetrating vehicle in both experimental and clinical studies. However, how TAT qualitatively or quantitatively affect the dynamics and distribution of its cargo is yet to be determined [20]. Several studies including ours suggest a fast brain accumulation with improved therapeutic effects, while other studies comparing brain delivery efficiency of five CPP vectors showed a moderate effect of TAT [21,22]. Most interestingly, one reported an astonishing 800 fold increase of ritonavir delivery into the CNS [23], suggesting the efficiency is not only determined by TAT but also cargo related.

Considering that MVIIA has typical side-effects of motor disorders and abnormalities in the nervous system, we also evaluate effects of MVIIA-c on the coordinated locomotion and tremor symptom. When the same dose of MVIIA and MVIIA-c (0.9 nmol/kg) were administered via intrathecal injection in the rotarod test, both groups of mice showed coordinated locomotion dysfunction. However, the severity level of the symptoms in the MVIIA group were more obvious than those of MVIIA-c group. Similarly in the tremor symptom test, MVIIA induced obviously more severe tremor symptom than MVIIA-c did at 30 min, although the tendency of tremor symptom at 120 min post-injection was following the order of MVIIA-c > MVIIA. The molecular basis of toxicity of N-type calcium channel inhibitor MVIIA has been studied in detail previously, and the key residues of MVIIA that contribute to its side effects have been clarified [24]. In theory, since we use the TAT modification strategy to achieve intranasally and/or systemically applicable MVIIA mutant peptides with reduced toxic effect, other analgesic toxin peptides with similar mechanisms can also be similarly modified to achieve analgesic effects by intranasal and/or systemic administration.

## 4. Materials and Methods

### 4.1. Peptide Synthesis

Crude peptides were all purchased from Beijing SciLight Biotechnology Ltd. Co.; glutathione (GSH), oxidized glutathione (GSSG), DTT, and cysteine were obtained from Gibco (Carlsbad, CA). All other reagents were analytically pure. All peptides were folded and purified as described previously [24,25]. The amino acid sequences of MVIIA and its variants are listed in Scheme 1.

### 4.2. Animals

Kunming mice (18–22 g, Beijing Animal Center, China) of both sexes were used. Male mice were used in acetic acid-induced writhing test, and female mice were used in hot-plate test. The mice were housed under a 12 h/12 h light/dark cycle at 23 ± 2 °C with a relative humidity of 50%. The mice had free access to food and water. All experiments were conducted in accordance with the guidelines of Animal Research Advisory Committee in Beijing Institutes for Biological Science. All procedures adhered to the Guidelines of the International Association for the Study of Pain regarding animal experimentation [26]. The protocol was approved by IACUC (Institutional Animal Care and Use Committee, Study Number QDU1801P, approval received on March 23, 2018).

### 4.3. I.C.V and I.N Injection Procedures

Brain intracerebral ventricular injection (I.C.V) and intranasal administration (I.N) were performed as previous reported [27]. Briefly, mice were anesthetized using 5% isoflurane (RWD Instrument, RWD Life Science Inc, San Diego, CA, USA) and placed in a stereotaxic frame. Mice skull was exposed and punctured through with a needle, the brain catheter was implanted into the skull at the following coordinates: 0.46 mm posterior to bregma, 1 mm lateral to the midline, and 2.2 mm ventral to the surface of the skull, fixed with two screws and acrylic cement. The incision was closed with sutures and the mice were returned to their home cage and housed individually. Experiments were performed 1 week after surgery. For I.C.V injection, a micro syringe filled with saline or peptide was connected by a polyethylene tube to a 30-gauge cannula, fitting into the guild cannula. Vehicle or peptide (1 μL) were injected into the lateral ventricles at the rate of 0.5 μL/min the needle was left in place for 2 min after infusion.

Intranasal treatment was performed according to the previous reports [28]. ICR mice were employed for this experiment. To facilitate the drug delivery with a micropipettor, mice restrained with the neck held parallel to the floor. Saline or peptide solution were administered via a micropipette tip inserted 4 mm into the left nostril of mouse.

### 4.4. CD Spectroscopy

The peptides were dissolved in phosphate buffer (10 mM, pH = 7.2) to a final concentration of 35 μM. The CD spectra were measured between 190 and 260 nm on a Chirascan Plus spectropolarimeter (Applied Photophysics Ltd., Leatherhead, Surrey, UK) at room temperature. The measurement parameters were set up as follows: step resolution 1.0 nm; speed 20 nm/min, and cell path length of 1.0 mm.

### 4.5. Electrophysiology

HEK293 cells expressing the SV40 large T-antigen (HEK293T) were cultured in Dulbecco’s Modified Eagle’s Medium (Gibco, Grand Island, NY, USA) supplemented with 10% (*v*/*v*) fetal bovine serum (FBS, Gibco), 100 units of penicillin, and 100 units of streptomycin at 37 °C in 5% CO_2_. The rat Ca_V_2.2 channel, α_1B_ splice variant e37a, auxiliary subunit α_2_δ_1_ and β_3_ were provided by Dr. Diane Lipscombe (Addgene plasmid # 26569, # 26575, # 26574). HEK293T cells were transiently co-transfected by Lipofectamine (Invitrogen, Carlsbad, CA, USA), as described previously [24], with 3 μg rat Ca_V_2.2 channel α_1B_, α_2_δ_1_ and β_3_ auxiliary subunits, and 0.4 μg enhanced green fluorescent protein. One day after transfection, cells were seeded onto glass chips and incubated at 37 °C in 5% CO_2_ for at least 6 hours before recording.

Whole-cell voltage clamp recording has been described previously [24]. Briefly, recording electrodes, with a resistance of ~3 MΩ, were filled with an internal solution containing the following (in mM): 135 CsCl, 10 NaCl, 10 HEPES, and 5 EGTA, pH 7.2, with CsOH. The extracellular recording solution contained (in mM): 135 *N*-Methyl-d-glucamine, 10 BaCl_2_.2H_2_O, 2 MgCl_2_.6H_2_O and 10 HEPES, pH 7.4. All currents were recorded at room temperature (~22 °C) with a MultiClamp 700B amplifier (Molecular Devices, Sunnyvale, CA, USA) controlled by a Clampex 10.3/Digidata 1440A acquisition system. Membrane currents were filtered at 2 kHz and sampled at 10 kHz. All data were analyzed with the clampfit 10.3 software (Molecular Devices), presented as mean ± SEM. All the dose-response curves of toxin blocking N-type ICa were obtained using GraphPad Prism (GraphPad Software version 5.01, San Diego, CA, USA) by plotting the inhibition of current amplitude as a function of drug concentration and were fitted using a hill equation.

### 4.6. Hot-Plate Test

Nine groups of 6–8 female mice were intracerebroventricularly I.C.V. administered MVIIA (0.11, 0.33 or 1.00 nmol/kg) or intravenously (I.V.) administered MVIIA or MVIIA-c (0.30, 1.00, or 3.00 μmol/kg) or intranasally (I.N.) administered MVIIA-c (3.3, 6.6, 9.9 nmol/kg). Saline was also administered in each ways as vehicle treated groups. The animals were placed on a hot plate with a constant temperature of 55 ± 0.5 °C. The latency time was recorded from the placement on the heated surface to the first licking of the hind paws or jumping as an index of pain threshold (Eddy and Leimbach, 1953). A cut-off time of 60 s was used to avoid tissue damage. The latency time was measured before administration as the baseline latency (0 h) and 0.5, 1, 2, 3, 4, 6, 8, 10, and 12 h after administration with MVIIA, MVIIA-c or Saline (I.C.V. or I.V.) respectively. Mice with baseline latency of less than 5 s or more than 20 s were eliminated to exclude hyposensitive or hypersensitive mice.

### 4.7. Acetic Acid-Induced WRITHING test

For the writhing test, mice were administered MVIIA (0.11, 0.33, or 1.00 nmol/kg, I.C.V.) or MVIIA-c (0.60, 1.80, 5.40 nmol/kg, I.C.V.), 30 min before injection of 1% acetic acid intraperitoneally (I.P), to check their analgesic activity in vivo as described previously [29]. To test the ability to penetrate the blood brain barrier, mice were administered MVIIA and MVIIA-c (0.30, 1.00, or 3.00 μmol/kg) intravenously, 3 hours before injection of 1% acetic acid (I.P). Saline was also administered as vehicle treated groups (I.C.V. and I.V.). The number of writhing responses was counted from 5 min to 20 min after acetic acid injection. The number of writhing movements was characterized by abdominal muscles contractions accompanied with stretching of hind limbs and elongation of the body. The percentage of inhibition was calculated using the following formula: Inhibition (%) = (N_0_− N_1_) × 100/N_0_, Where N_0_ is the writhe number of Saline, and N_1_ is the writhe number of experiments.

### 4.8. Coordinated Locomotion in Rotarod Test

In this study, rotarod test were performed to evaluate the motor impairment of MVIIA (0.9 nmol/kg) or MVIIA-c (0.9 nmol/kg) as described in a previous study with some modifications [24]. Before the test, animals were trained at the speed of 10 rpm for 3 min once a day for three days. The peptides or saline were administered I.C.V. to the mice (n = 8–10 in each group, half females and half males) in a volume of 6 μL. At 30 and 120 min post-injection, the mice were placed on the rotating rod at a speed of 5 rpm for 68 s. Subsequently, the rod was accelerated from 5 to 30 rpm over a 3 min interval. In the acceleration process, the time during which the mice remained balanced on the rotating rod was recorded (3 min as the maximum time).

### 4.9. Tremor Symptom Test

The tremor period was regarded as the time spent by the mouse having rhythmic oscillatory movements of the limbs, head, and trunk induced by peptides as previously described [30]. The mice were randomly divided into three groups: MVIIA (0.9 nmol/kg) group, MVIIA-c (0.9 nmol/kg) group and a control group (6 μL, I.C.V.; n = 12, half females and half males). After 30 and 120 min, the accumulative tremor time (s) in the period of 5 min was recorded by a digital camera and scored by a blinded observer.

### 4.10. Statistical Analysis

The results of analgesic and toxicological tests were analyzed using one-way of variance (ANOVA), two-analysis of variance with repeated measures, followed by the Student-Newman-Keul’s test. All data are presented as mean ± stand error (SEM). The differences with p values less than 0.05 were considered statistically significant.

## 5. Conclusions

Taken together, our results showed that the naturally derived analgesic peptide MVIIA was rendered BBB permeability after fusion with TAT at its C-terminus. This conjugation, although reduced the binding efficacy, changes the time response character of MVIIA. More interestingly, MVIIA-c exhibited a prolonged time response window in analgesic efficacy with apparently reduced side effects, which may provide therapeutic benefits in various different clinical scenarios of pain management. Our results that intranasal administration of MVIIA-c exhibited strong analgesic effects comparable to that of I.V. and I.C.V administrations could provide an extra feasibility for clinical application, not only because of the lower doses required, but also because of fewer side effects.

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
