# Peer review of "TAT-Modified ω-Conotoxin MVIIA for Crossing the Blood-Brain Barrier"

_marinedrugs, 2019, doi:10.3390/md17050286_

Reviewer 1 Report

Major

Why were not all MCIIA-TAT variants tested for their ability to cross the blood-brain barrier? Even though MVIIA-c displayed the lowest IC50 value, all variants display a very similar activity. Actually, all IC50 values are within a 2-fold magnitude of each other. It might very well be that some of the other variants (MVIIA-a, MVIIA-b & MvIIA-d) display a better BBB-penetrating ability and thus show better results than MVII-A. The antinociceptive experiments needs to be performed with the other variants. Without these results, it is not convincing that MVIIA-c is the best choice for further therapeutic design. 

Line 112: Electrophysiology results: Current traces of all tested peptides at their IC50 values needs to be provided.

Figure 2:

-  It is interesting that the dose-response curve of MVIIA displays a H = 1.296 while this of MVIIA-c has a H value of 2.387. How do the authors explain this? This should be discussed in the manuscript.

- The fits in panel A, B, C and D needs to be corrected. Now it seems that the dose-response curve never reaches 0 value. This is important because the authors made the choice to continue only with variant MVIIA-c, solely based on the IC50 values obtained. However, the corrected fits might show some different IC50 values.

-  Panel F: It seems that the labelling of 10 µM L-MVIIA is switched with this of 2 µM MVIIA? According to the text 10 µM L-MVIIA showed an inhibition of ± 20% while in the figure F it completely blocks the Ca2+ current?

Minor

Line 37: Introduction on ziconotide should also mention the side effects observed upon use of this drug.

Line 55: … administration and…

Line 56: The reference of Murthy et al is not included in the reference list.

Line 113: the conventional way to write CaV2.2 is with the V in subscript. This needs to be corrected throughout the manuscript.

Line 140: … shed the…

Author Response

Why were not all MCIIA-TAT variants tested for their ability to cross the blood-brain barrier? Even though MVIIA-c displayed the lowest IC50 value, all variants display a very similar activity. Actually, all IC50 values are within a 2-fold magnitude of each other. It might very well be that some of the other variants (MVIIA-a, MVIIA-b & MvIIA-d) display a better BBB-penetrating ability and thus show better results than MVII-A. The antinociceptive experiments needs to be performed with the other variants. Without these results, it is not convincing that MVIIA-c is the best choice for further therapeutic design.

Answer: The main objective of this paper is to test whether the fusion protein strategy with TAT peptide could render MVIIA BBB penetration capacity, and intravenous administration  could produces analgesic effects in animals. Considering most MVIIA-TAT variants exhibit similar electrophysiological activity, and folding and purifying each peptide and evaluating its animal activity require a lot of work, we only choose MVIIA-c for further study. We acknowledge that the animal activity of these MVIIA-TAT variants may not be completely consistent with the cell activity, but our experimental objectives have been achieved, and find peripheral administration MVIIA-TAT maintains analgesic efficiency with prolonged effective time window.

Line 112: Electrophysiology results: Current traces of all tested peptides at their IC50 values needs to be provided.

Answer: We have revised Figure 2, the current traces of peptides have been added.

Figure 2:

It is interesting that the dose-response curve of MVIIA displays a H = 1.296 while this of MVIIA-c has a H value of 2.387. How do the authors explain this? This should be discussed in the manuscript.

Answer: We discussed it in the manuscript.

“Interestingly, the hill slope increases with the increase of  linker length, when linker is three amino acids in length the hill slope reaches to the max(2.387), while the hill slope of MVIIA is only 1.296. After that, the hill slope decreases again when the linker length continue grow. As is known to us, hill slope reflects the binding situation of ligand and receptor. Since the linkage between inhibitor binding and response can be very complex, the mechanism of the change of hill slope in this case needs considerate in further studies.”

The fits in panel A, B, C and D needs to be corrected. Now it seems that the dose-response curve never reaches 0 value. This is important because the authors made the choice to continue only with variant MVIIA-c, solely based on the IC50 values obtained. However, the corrected fits might show some different IC50 values.

Answer: We normalized all the data, did the nonlinear regression again and recalculated IC50 of all the peptides in Graphpad Prism 5.01. We find out that all the dose-response curves can reach 0 value, but Hill slopes or IC50 values did not change at all.

Panel F: It seems that the labelling of 10 µM L-MVIIA is switched with this of 2 µM MVIIA? According to the text 10 µM L-MVIIA showed an inhibition of ± 20% while in the figure F it completely blocks the Ca2+ current?

Answer: We made a mistake when drawing the legend. The mistake has been corrected.

Line 37: Introduction on ziconotide should also mention the side effects observed upon use of this drug.

Answer: We have mentioned the ziconotide side effects at section 2.5 (Effects of peptides on the coordinated locomotion), section 2.6 (Tremor symptom in mice induced by peptides) and Line 265 (In addition, systemic administration of ziconotide is also known to cause profound side effects [25]). But considering the style of writing, we hope that the introduction will lead to the scientific problems we need to solve.

Line 55: … administration and…

Answer: The mistake has been corrected.

Line 56: The reference of Murthy et al is not included in the reference list.

Answer: The reference of Murthy et al is No 5 in the reference list, and one more indexes has been added at the end of sentence.

Line 113: the conventional way to write CaV2.2 is with the V in subscript. This needs to be corrected throughout the manuscript.

Answer: The mistake has been corrected.

Line 140: … shed the…

Answer: The mistake has been corrected.

Reviewer 2 Report

The manuscript describes the analysis of MVIIA mutants linked to the TAT peptide. The results appear to be interesting with in vivo effects of one of the mutant peptides when administered iv. The major issue I have with the study is the analysis of the structures of the peptides. For instance, it is not clear if the additional residues added to the N-terminal of MVIIA influence the bioactivity directly or if the peptide has a different structure and disulfide connectivity. It might be necessary to use a range of oxidation conditions to obtain the native disulfide connectivity.

As stated by the authors the CD data is difficult to interpret. Analysis of the peptides with NMR spectroscopy would be a straight forward was to assess the structural influences of the N-terminal and C-terminal extensions. Chemical shift analysis is likely to indicate if the overall fold is maintained and provide an indication of whether the native disulfide connectivity is present. In the absence of NMR analysis, a chemical analysis of the disulfide connectivity would be required to provide insight into the structure/function relationships. This insight would significantly improve the study.

Author Response

The manuscript describes the analysis of MVIIA mutants linked to the TAT peptide. The results appear to be interesting with in vivo effects of one of the mutant peptides when administered iv. The major issue I have with the study is the analysis of the structures of the peptides. For instance, it is not clear if the additional residues added to the N-terminal of MVIIA influence the bioactivity directly or if the peptide has a different structure and disulfide connectivity. It might be necessary to use a range of oxidation conditions to obtain the native disulfide connectivity.

As stated by the authors the CD data is difficult to interpret. Analysis of the peptides with NMR spectroscopy would be a straight forward was to assess the structural influences of the N-terminal and C-terminal extensions. Chemical shift analysis is likely to indicate if the overall fold is maintained and provide an indication of whether the native disulfide connectivity is present. In the absence of NMR analysis, a chemical analysis of the disulfide connectivity would be required to provide insight into the structure/function relationships. This insight would significantly improve the study.

Answer: Structural analysis of peptides has always been a challenging task. In this paper, circular dichroism is used to characterize peptides, but little information is revealed. However, considering that MVIIA-a-d has the similar electrophysiological activity with MVIIA, they can be considered to have the same disulfide bond pairing mode. Analysis of the peptides with NMR spectroscopy would be a straight forward was to assess the structural influences, next, we will work with nuclear magnetic experts to study the structure of peptides.

Round  2

Reviewer 1 Report

The authors have addressed all my concerns adequately. I have no further comments.

Author Response

The authors have addressed all my concerns adequately. I have no further comments.

Answer: Thank you very much.

Reviewer 2 Report

Although it is certainly possible that the peptides have the same disulfide bond pairing as the native peptide, the peptides still display 10 fold less activity and there are examples of non-native peptides with some level of bioactivity. Many of the conclusions related to structure/function relationships are based on the disulfide connectivity being maintained.  If the connectivity is not going to be elucidated using NMR or chemical methods, I think it should be clearly stated that this could be a reason for the differences.

Author Response

Although it is certainly possible that the peptides have the same disulfide bond pairing as the native peptide, the peptides still display 10 fold less activity and there are examples of non-native peptides with some level of bioactivity. Many of the conclusions related to structure/function relationships are based on the disulfide connectivity being maintained.  If the connectivity is not going to be elucidated using NMR or chemical methods, I think it should be clearly stated that this could be a reason for the differences.

Answer: We have added some sentences in the lines 241-244 of the manuscript.

"Although it is certainly possible that the MVIIA derivatives have the same disulfide bond pairing as MVIIA, the derivatives still display 5-10 fold less activity. This may also be caused by disulfide bond mismatch. Next, we will study the disulfide bond pairing of derivatives by digestion and mass spectrometry."

Mar. Drugs EISSN 1660-3397 Published by MDPI AG, Basel, Switzerland RSS E-Mail Table of Contents Alert
Back to Top